# Matching Assistive Technology, Telerehabilitation, and Virtual Reality to Promote Cognitive Rehabilitation and Communication Skills in Neurological Populations: A Perspective Proposal

**Fabrizio Stasolla** [1,*] **, Antonella Lopez** [1,2] **, Khalida Akbar** [3] **, Leonarda Anna Vinci** [1] **and Maria Cusano** [1]

1   Law Department, "Giustino Fortunato" University of Benevento, 82100 Benevento, Italy
2   Department of Educational Sciences, Psychology, Communication, Università degli Studi di Bari Aldo Moro, 70121 Bari, Italy
3   Faculty of Management Sciences, Durban University of Technology, Durban 4041, South Africa
*   Correspondence: f.stasolla@unifortunato.eu

**Abstract:** Neurological populations (NP) commonly experience several impairments. Beside motor and sensorial delays, communication and intellectual disabilities are included. The COVID-19 pandemic has suddenly exacerbated their clinical conditions due to lockdown, quarantine, and social distancing preventive measures. Healthcare services unavailability has negatively impacted NP clinical conditions, partially mitigated by vaccine diffusion. One way to overcome this issue is the use of technology-aided interventions for both assessment and rehabilitative purposes. Assistive technology-based interventions, telerehabilitation, and virtual reality setups have been widely adopted to help individuals with neurological damages or injuries. Nevertheless, to the best of our knowledge, their matching (i.e., combination or integration) has rarely been investigated. The main objectives of the current position paper were (a) to provide the reader with a perspective proposal on the matching of the three aforementioned technological solutions, (b) to outline a concise background on the use of technology-aided solutions, (c) to argue on the effectiveness and the suitability of technology-mediated programs, and (d) to postulate an integrative proposal to support cognitive rehabilitation including assistive technology, telerehabilitation, and virtual reality. Practical implications for both research and practice are critically discussed.

**Keywords:** neurological populations; technology-based interventions; quality of life; inclusion; cognitive rehabilitation; practical issues

## 1. Introduction

The term neurological population (NP) usually refers to any individual who presents an acquired or congenital damage and/or injury to the central nervous system. The autonomic nervous system, brain, cranial nerves, muscles, nerve roots, neuromuscular junctions, peripherical nerves, and spinal cord are usually embedded. Persons with neurological impairments may have different levels of intellectual disabilities, sensorial disorders, motor delays, communication inabilities, and lack of speech. People affected by neurodevelopmental disorders (i.e., attention deficits hyperactivity disorders, autism spectrum disorders, cerebral palsy, rare genetic diseases), neurodegenerative diseases (i.e., Alzheimer's and Parkinson's disease, lateral amyotrophic sclerosis, and multiple sclerosis), and/or with acquired brain injuries and stroke, and post-coma patients, either in a vegetative state or in a minimally conscious state, may be included.

The main feature of these categories of patients is the presence of multiple disabilities (ranging from mild–moderate to severe or profound) that has consequences on the level of residual functioning [1–5]. Therefore, persons with neurological impairments necessarily rely on caregivers, families, practitioners, and professionals' daily assistance [6,7]. Because they are unable to positively tackle daily environmental requests, persons diagnosed with

neurological impairments may experience detachment, isolation, passivity, and withdrawal throughout their lifespan as a chronic clinical condition [8–10]. Whenever an acute clinical condition occurs, it may seriously hamper their social image and status [11,12]. Thus, it may be deleterious for their quality of life. In fact, their clinical conditions may have negative outcomes on caregivers, professionals, and families' burden [13–15].

AT includes devices, equipment, pieces, or tools capable of ensuring NP independence and self-determination. Based on learning principles (i.e., causal association between behavioral responses and environmental consequences), an AT-based program provides a functional bridge between the limited human repertoire and high environmental requests. Accordingly, an active role, constructive engagement, positive participation, and profitable occupation will be enhanced [16–22]. Microswitches represent a basic form of AT-based interventions and usually include electronic sensors and tools capable of detecting small behavioral responses. Consequently, those tools may provide NP affected by multiple disabilities with brief periods of positive stimulation. Vocal output communication aids and/or speech-generating devices may enable NP to have social interactions mediated by a caregiver. Computer-mediated programs may ensure requests and choices of desired items with leisure and occupation options [23].

Recently, new technologies have rapidly been developed. VR enables ecological validity, behavioral tracking, experimental control, and immersive environments similar to those of real life. Serious Games promotes positive interactions and active participation. TR as part of telemedicine, offers the assessment, rehabilitation, and supervision of patients remotely, either in a synchronous or in an asynchronous modality. Robot-assisted interventions may be implemented in clinical or medical settings and may be used to rehabilitate patients with stroke. Both assessment and rehabilitative purposes may be favorably achieved [24,25]. Nevertheless, sporadic contributions have been published combining those technological solutions in cognitive rehabilitation (CR) [26,27].

CR is broadly defined as a systematic and oriented cognitive therapy focused on achieving and/or pursuing functional modifications by (a) restoring or fostering previous acquired patterns of behavior or (b) teaching or empowering new patterns of cognitive activity and compensatory mechanisms for impaired neurological systems [28]. Accordingly, CR may include either restorative or compensatory approaches. Restorative strategies are based on exercise principles. The repetitive exercise of neural connectivity supporting cognitive function will enhance the learning process of new skills, which will restore the damage caused by the injury. Restorative approaches rely on neuroplasticity. Thus, intact or nonimpaired neurons and neural circuits would replace the lost functions. Conversely, compensative strategies rely on the substitution of the neural pathways previously necessary to achieve a specific task. By replacing functional mechanisms, modalities are acquired to achieve specific goals. Both approaches are probably combined in a rehabilitative intervention because the cognitive processes stimulated during a compensatory training can easily enhance and consolidate neural connectivity or new learning [29,30].

Considering the above, the aim of the present position paper is to propose a review of the literature about technology-aided rehabilitative programs in NP, emphasizing how technologies can be helpful to overcome neurological insults. Moreover, attention is paid to the effectiveness and the suitability of technology-based interventions for communication and leisure skills. Finally, a new perspective proposal on the combination and matching of AT, TR, and VR in NP is presented.

## 2. Cognitive Deficits and Technological Supports

Patients with neurological injuries are commonly recognized to be affected by cognitive deficits (e.g., limited executive functions), communication disorders (e.g., lack of speech), motor impairments (e.g., gait or locomotion incapacities), and/or sensorial abnormalities (e.g., hearing or vision loss). Those impairments may negatively impact patients' daily functioning, causing an increased burden on either caregivers or professionals. Although standard measures such as cognitive scales have been largely adopted in clinical settings,

relevant limitations are frequently outlined. Thus, the patient's assessment requires intensive work, result grading has poor resolution (i.e., mild/moderate/severe/profound), and there is no interactive feedback based on large datasets. Furthermore, for testers' data to be considered reliable, intensive training would be mandatory. To tackle this issue, one may consider eye tracking, quantifying significant parameters such as amplitude, latency, frequency, and stability which capture objective and reliable dependent variables. Large datasets can be collected with sophisticated methods based on machine learning. Increasing evidence suggests that eye tracking information highly correlates with standard cognitive assessment scales, strongly supporting the idea that eye tracking can be used to evaluate cognitive states, disease severity and progression in NP [31].

AT, VR, and TR may be useful for both assessment and rehabilitative purposes because they provide a strong and valid support to evaluate and rehabilitate the cognitive functioning. For example, AT may be adopted in post-coma patients with disorders of consciousness to evaluate if a diagnosis of vegetative state is reliable or a more favorable diagnosis of minimally conscious state can be made [32]. Additionally, VR may be implemented as a smart aging platform for assessing early stages of cognitive impairments in patients with neurodegenerative diseases [33]. Moreover, TR may be used to positively supervise patients with neurological impairments remotely with a dual objective, namely, (a) evaluation and (b) rehabilitation [34,35].

### 3. Method and Selective Review

A computerized search was performed in SCOPUS. Neurological Populations, neurological impairments, neurodevelopmental disorders, ADHD, ASD, cerebral palsy, neurodegenerative diseases, acquired brain injuries, spinal cord injury, stroke, assessment, recovery, rehabilitation, AT, VR, TR, multiple sclerosis, amyotrophic lateral sclerosis, Alzheimer's and Parkinson's disease, post-coma, vegetative state, and minimally conscious state were merged as keywords. A manual search was additionally included of the published literature. The eligibility criteria were (a) an empirical contribution with a technology-based program, (b) at least a participant with neurological impairments, (c) last five years (i.e., 2018–2022 as the range interval of publication), and (d) English as the language of the paper. Theoretical papers, book chapters, conference papers, and/or proceedings were excluded since their inclusion would exceed the purpose of the current article.

Similarly, we did neither a meta-analysis nor a systematic review and we acknowledged this specific point as a limitation of our work (see the final section). Initially, 80 records were identified. Duplicates were preliminarily removed. Two scholars with a 10 year professional expertise on technology-aided interventions and multiple disabilities independently assessed and screened the selected contributions. First, a title assessment was conducted, and 37 contributions were retained. Second, an abstract evaluation was carried out, with 28 contributions screened. Third, a final decision based on the aforementioned eligibility criteria was made on 15 contributions reviewed. Whenever a disagreement occurred, a third scholar was involved to take a final decision. An interrater agreement of 97% was finally recorded. Initially, overall, the 15 papers reviewed (i.e., 5for each identified group, namely, AT, VR, and TR) [36–50] were grouped according to the technology used (i.e., AT, VR, or TR). Among the reviewed studies, five were based on single-subject experimental designs [36,44,45,49,50], four were between-groups experimental comparisons [38–40,47], three were randomized controlled trials [37,42,46], one included an experimental training [48], one was a prospective cross-sectional study [43], and finally one was a protocol with a transcranial deep stimulation [41]. The review process is available upon request to the corresponding author. For practical reasons, the contribution of each group is detailed. Table 1 summarizes the reviewed studies arranged in alphabetic order.

**Table 1.** Reviewed studies arranged in alphabetical order.

| Authors | Participants | Ages | Group | Technology | Main Outcomes |
|---|---|---|---|---|---|
| Arroyo-Ferrer et al. [36] | 1 | 20 | VR | EEG neurofeedback | Benefits in divided and sustained attention |
| Bekkers et al. [37] | 121 | 60–90 | VR | Virtual treadmill | Postural stability |
| Bertomeu-Motos et al. [38] | 8 | 22–58 | AT | Multimodal interface | Positive achievement of cognitive tasks in the experimental group |
| Calabrò et al. [39] | 22 | 60–73 | VR | Computer-assisted | Gait stability |
| Capodieci et al. [40] | 42 | 5–11 | TR | Computerized training | Accuracy in dictation, reading, inhibition, and working memory test |
| Eilam-Stock et al. [41] | 1 | 29 | TR | Deep stimulation | Improvement in attention and working memory |
| Gerber et al. [42] | 15 | 43–63 | TR | Interface with Hierarchical Structure | Participants' Enjoyment and Approval |
| Jamali et al. [43] | 43 | 4–12 | TR | Coaching technology | Improvement in occupational performance |
| Jordan et al. [44] | 41 | 9–13 | AT | Keyboard emulator | Literacy access and words prediction |
| Lancioni et al. [45] | 6 | 38–59 | AT | Smartphone and adapted software | Communication and leisure opportunities |
| Leonardi et al. [46] | 30 | 50–65 | VR | Rehabilitation system | Mood Improvement Visuo-spatial skills enhancement |
| Maier et al. [47] | 30 | 45–75 | VR | Adaptive conjunctive cognitive training | Improvement in attention, spatial awareness, and cognitive functions |
| Pinter et al. [48] | 14 | 32–40 | TR | EEG neurofeedback | Cognitive improvement correlated with increased functional connectivity |
| Stasolla et al. [49] | 10 | 7–10 | AT | Microswitches | Improvement of adaptive skills and positive participation |
| Stasolla et al. [50] | 5 | 14–18 | AT | Microswitches | Enhancement of academic performance and personal needs communication |

Bertomeu-Motos et al. [38] assessed the environment control interface (ECI) developed under a multimodal interface able to analyze and extract relevant information from the environment as well as from the identification of residual skills, behaviors, and intentions of the user (i.e., AIDE). Eight adults aged between 22 and 58 years with different neurological diseases such as muscular atrophy, dystrophy, ischemic stroke, and spinal cord injury participated in a simulated scenario using a two-screen layout: one with the ECI, and the other with a simulated environment, developed for this specific purpose. The sensorimotor rhythm and the horizontal oculo-version were used to measure the online monitoring of the ECI after the user training and system calibration. The participants were requested to perform simulated tasks consisting of daily living actions, such as drinking, switching on a lamp, or raising the bed head, for ten minutes. Two experimental conditions were evaluated, namely, (a) the AIDE mode, using a prediction model useful to recognize the user intention facilitating the scan, and (b) the Manual mode, without a prediction model. The results evidenced that the mean task time spent in the AIDE mode was shorter if compared to that of the manual mode. Thus, the participants positively achieved more tasks in the AIDE mode with statistical differences between conditions. Additionally, all the participants correctly performed 90% of the activities using the AIDE mode. Conversely, at least three steps were necessary in the manual mode.

Leonardi et al. [45] evaluated cognitive outcomes after rehabilitation training mediated by the VR rehabilitation system (VRRS) in patients diagnosed with multiple sclerosis (MS). Thirty participants with relapsing/remitting MS were enrolled. They were aged between 50 and 65 years. All the participants were equally and randomly assigned either to a

control group (CG) or to an experimental group (EG). The CG received traditional cognitive rehabilitation training. The EG was exposed to a VR-based intervention. Both groups underwent an identical amount of cognitive rehabilitation, three times per week, for 8 weeks. They were assessed through neuropsychological evaluation before (T0) and after (T1) the rehabilitation program. The data demonstrated that both rehabilitative approaches improved the participants' mood and visuospatial skills. Nevertheless, only the EG showed a significant improvement in specific cognitive domains such as learning abilities, short-term verbal memory, lexical access ability, and quality of life related to mental states.

Eilam-Stock, George, and Charvet [41] reported a case study of a 29-year-old man with traumatic brain injury (TBI) with persisting negative consequences on both cognitive and emotional domains. A remote transcranial direct current stimulation (tDCS) paired with cognitive training was supplied. Neuropsychological measures were recorded before and after the participant completed a set of 20 daily sessions of remote supervision (RS.tDCS; 2.0 mA × 20 min, left anodal dorsolateral prefrontal cortex montage). During the stimulation period, the participant completed adaptive cognitive training. All the procedures were implemented at home and monitored remotely in real time through videoconferences with a study technician. The results showed a significant enhancement and empowerment on tests of attention and working memory, semantic fluency and information processing speed. The participant's mood improved as well. The contribution empirically corroborated the effectiveness and the suitability of an RS-tDCS-based program to improve cognitive skills following a TBI.

## 4. Communication, Internet, and Leisure Opportunities

Stroke, traumatic brain injuries, cerebral palsy, congenital conditions, and/or neurodegenerative diseases may cause negative communication outcomes such as apraxia of speech, aphasia, dysarthria, lack of speech. In fact, individuals with neurological impairments may experience systematic failures in communication skills, with poor social interactions and limited opportunities to positively satisfy their personal needs [51–53]. Accordingly, persons with neurological disorders may be unable to easily understand what is being said or told. Additionally, they may be isolated and passive, with negative consequences on their quality of life and increased caregivers' burden [54,55]. Thus, restricted education and/or employment opportunities are acknowledged. Usually, a traditional speech-mediated intervention with a therapist is recommended [56]. Rehabilitative programs are commonly delivered within a medical setting. That is, limited generalized strategies and learning processes may be recognized [57]. Technology-aided interventions may be considered highly encouraging and promising to overcome this issue [58,59].

For example, one may envisage the use of speech-generating devices to promote requesting behavior. Specific hardware (e.g., tablets, IPAD, or IPOD) with adapted software capable of capturing the participants' needs may be adopted [60,61]. Both neurodevelopmental disorders and neurodegenerative diseases may be targeted [62,63]. Furthermore, one may use computerized systems with adapted software to support requests and choice behaviors among different options organized in a hierarchical way [59,60]. Furthermore, smartphones and microswitches to support the self-management of telephone calls and short text messages may be used [64,65]. Finally, one might consider internet access through technological aids supporting leisure opportunities [66,67].

VR setups within emerging technologies have recently attracted relevant attention and increasingly become affordable and accessible to be implemented in clinical settings. Additionally, those programs may represent a valid tool for both assessment and recovery of cognitive functioning [68,69]. Communication-based interventions may be fostered. Thus, ecological validity, experimental control, and behavioral tracking as mentioned above may be ensured with real-word immersive and/or non-immersive environments [70–72]. Nevertheless, few studies have been conducted, up to date, on the enhancement of communication skills in NP through VR setups [73,74]. Finally, TR has the potential to enable high-frequency treatment remotely. An illustrative example is outlined by Uslu et al. [75]

who investigated the effectiveness and the suitability of a new tablet-based telerehabilitation speech and language therapy-based app in patients with aphasia. A randomized, controlled, evaluator-blinded, multicenter trial protocol was formulated. A systematic comparison with a tele-rehabilitative cognitive training was carried out. A sample of 100 patients with aphasia was considered. The patients were assigned to two groups in a 1:1 ratio stratified by trial, site, and severity of impairment. Both groups were trained over a period of 4 weeks for 2 h per day. The experimental group devoted 80% of the training to the new app managed through a tablet and 20% of the time to the cognitive training; the treatment percentages were reversed for the control group. The primary outcome included an understandable verbal communication. The secondary outcome included the intelligibility of the verbal communication, the impairment of both expressive and receptive communication, and confrontation naming. Further outcomes measures were the quality of life and acceptance (i.e., the usability of the system combined with the subjective experience).

## 5. The New Perspective Proposal

In the light of the above, the possibility to combine and match technology-based setups using AT, TR, and VR devices and tools for enabling the reliable assessment and recovery of cognitive functioning in NP [76,77] could be considered useful. The novelty feature may include a unique combined technology-aided strategy helpful to promote an active role, constructive engagement, and positive participation in NP. Individuals with neurological impairments may be involved in immersive daily situations similar to those of the real life. The mediation of both families and caregivers might be fostered. The remotely supervised assessment and recovery of cognitive functions might be pursued. Customized solutions, tailored options, and individualized devices, equipment, or tools might be envisaged. An integrated and combined technology-mediated program can profitably be built to support NP and their families and caregivers in daily life.

The implementation of this proposal could improve the patient's quality of life in preventing isolation and passivity and reduce both caregivers and families' burden [78]. Specifically, it is conceived for matching the three approaches in a unique suitable setup to be implemented in clinical settings [79]. The TR strategy might additionally be used remotely while the patient is living at home [80]. Systematic comparisons between healthy individuals and matched NP may be sought [81]. Basic AT devices may fill the existing gap between an individual's behavioral repertoire and environmental requests [82]. VR-based programs rely on fully immersive illusions. A first-person centered perspective is usually considered. A customized solution is commonly adopted [83]. TR should ensure remote monitoring and supervision in a synchronous or asynchronous modality [84,85].

Although no specific rules exist, the following steps may be emphasized to maximize and optimize the learning process. First, a suitable behavioral response available in the individual's repertoire should be identified. Second, a suitable technological setup capable of detecting the behavioral response and suitable for delivering environmental consequences should be implemented. Third, highly motivating, pleasant, and rewarding events to be used as positive reinforcers should be selected through a formal screening [86]. Whenever those guidelines are applied, one may reasonably argue that the learning process will be successfully fostered [87].

The proposed matching between technological solutions may represent a valid alternative to overcome the current COVID-19 pandemic situation, which worldwide requires effective and customized technological options [88,89]. Next to healthcare and well-being during the COVID-19 pandemic era [90,91], a growing interest has recently been devoted to NP. Some illustrative and practical examples are presented. For instance, one may rely on the application of serious games among children and adolescents with neurodevelopmental disorders with both fun and educational purposes [92,93]. Otherwise, patients with acquired brain injuries and post-coma conditions might be assessed and helped recover through technological-aided programs [94]. Additionally, participants with neurodegenera-

tive diseases may be enabled to communicate with distant partners via technology-aided interventions [95]. Accordingly, positive outcomes in an individual's health and well-being may be enhanced. Caregivers and families' burden may be significantly reduced [96–98]. A full social inclusion of NP in daily settings may be fostered [99,100]. The combination should be highly customized and rigorously tailored on the user and the targeted cognitive impairments investigated. Depending on the participant's level of functioning, the diagnosis and capacity/possibility of recovery, different combinations might be envisaged. For example, for individuals with multiple disabilities and a low behavioral repertoire, assistive technology-based devices integrated with serious games can be assessed. For patients with neurodegenerative diseases and cognitive decline, VR setups and telerehabilitation can be merged. For patients with post-coma and disorders of consciousness, based on their level of functioning, an assistive technology device combined with a TR service could be considered. Systematic empirical investigations/studies in this regard to evaluate each specific situation are mandatory.

## 6. Discussion

The literature available on the use of technological-aided interventions in individuals affected by neurological conditions emphasizes the suitability and the effectiveness of such programs for both assessment and recovery purposes [101–103]. AT profitably ensures a functional bridge between the behavioral repertoire and environmental requests enabling people to acquire independence and self-determination [104,105]. VR positively presents the real word with immersive and/or non-immersive situations with a fully sensorial pleasant experience [106,107]. TR helps clinicians and practitioners in remote assessment and cognitive daily rehabilitation [108,109].

Unfortunately, no empirical evidence exists on the effects of using combined and matched technologies in a unique program. Matching the technological solutions may provide NP with (a) a further active role, (b) an advantageous and supportive constructive engagement, (c) immersive real-word conditions with fully sensorial experience, and (d) remote assessment and rehabilitation. Executive functions (e.g., sustained attention and working memory) and communication skills (i.e., verbal and written) may be relevantly promoted. The affordability, accessibility, effectiveness, and suitability in daily settings of the combined strategy should be carefully and rigorously investigated through empirical and systematic contributions involving different neurological conditions. The sustainability with regard to (a) financial resources, (b) human resources, and (c) environmental availability should be also evaluated, as well as the targeted behaviors (e.g., communication skills, challenging behaviors, social and/or emotional behaviors, executive functions) [110–113].

Indeed, leisure, positive participation, functional occupation, sustained attention, and communication may be significantly enhanced. A purposeful behavior with an active role might be additionally documented [114,115]. Negative outcomes in the participants' mood such as depression and passivity may be meaningfully prevented [116,117]. The helpful use of AT and combined emerging technologies in NP should be empirically corroborated [118,119]. VR setups were useful to promote positive engagement [120,121]. TR provided remote monitoring and recovery [122]. Because the COVID-19 pandemic may be deleterious and may seriously hamper the independence and self-determination, an active role, constructive engagement, and positive participation of NP in daily activities should primarily be considered. To overcome negative consequences in NP quality of life and caregivers' burden, combined and integrated technology-aided solutions may be viewed as highly recommended [123]. Matching the three strategies through a careful and customized selection of devices and equipment may be highly warranted and may provide NP with a full, valuable, and complete support to tackle daily issues [124].

## 7. Limitations and Future Directions

Although relevant, encouraging, and promising, our perspective proposal has some limitations. First, it lacks an empirical demonstration. Second, we did not adequately

differentiate the neurological conditions (e.g., neurodevelopmental disorders and neurodegenerative diseases). Third, neither a systematic review nor a meta-analysis is present in the available literature. Rather, a selective review was conducted.

Beside supporting our proposal with empirical studies, future directions for both research and clinical practice should deal with the following topics: (a) systematic comparisons between different technological solutions, devices, or tools, (b) generalization, maintenance, and follow-up experimental phases, (c) extension to new technology-based strategies which should be systematically customer-tailored, and (d) preference checks or social validation procedures including external raters (e.g., caregivers, neurologists, physiotherapists, or psychologists) such as professional experts to support the clinical validity [125].

## 8. Conclusions

Neurological populations (NP) commonly experience impairments such as motor and sensory delays and communication and intellectual disabilities, and the COVID-19 pandemic has worsened their clinical conditions. The compatibility of technology-assisted interventions, such as assistive technology-based interventions, telerehabilitation, and virtual reality, has rarely been investigated. This paper provides a perspective proposal on the pairing of the three previously mentioned technological solutions, outlines a concise background on the use of technology-assisted solutions, argues on the effectiveness and suitability of technology-mediated programs, and presents an integrative proposal to support cognitive rehabilitation.

**Author Contributions:** Author Contributions. F.S. conceived and drafted the paper. A.L. and K.A. edited and critically revised the manuscript. L.A.V. and M.C. revised the paper. All authors have read and agreed to the published version of the manuscript.

**Funding:** This research received no external funding.

**Institutional Review Board Statement:** Not applicable.

**Informed Consent Statement:** Not applicable.

**Data Availability Statement:** The data presented in this study are available on request from the corresponding author.

**Conflicts of Interest:** The authors declare no conflict of interest.

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
