# Peer review of "Matching Assistive Technology, Telerehabilitation, and Virtual Reality to Promote Cognitive Rehabilitation and Communication Skills in Neurological Populations: A Perspective Proposal"

_technologies, doi:10.3390/technologies11020043_

Round 1
Reviewer 1 Report
1. A brief summary:
The main objectives of the paper were to provide the reader with a perspective proposal on the matching of assistive technology-based interventions, telerehabilitation, and virtual reality; to outline a concise background on the use of technology-aided solutions; to argue on the effectiveness and the suitability of technology-mediated programs; and to postulate an integrative proposal to support cognitive rehabilitation including the three mentioned solutions.
2. General concept comments:
a) Weaknesses of the paper: more theoretical paper but as for opinion paper it is acceptable
b) Strengths of the paper: an important and up-to-date topic, refers to modern technologies, shows new possibilities
c) Hypotheses / goals / research gap: well-set, clearly presented goals, there should be more about the research gap
d) Methodology: accurate
e) Literature: 125 references, almost every line is supported by the literature, proper selection
3. Specific comments:
· Please add more information about the research gap.
· 198 and 204 lines – there is no need for such a space.
· There is a discussion, but I recommend adding conclusions – it is good that the authors presented limitations and future directions however there should be a separated chapter named Conclusions before or after anyway. Adding conclusions will be clearer for readers to figure out the most important fundings of the study.
· The manuscript is clear, relevant for the field, presented in a well-structured manner.
Author Response
Dear Reviewer:
Thank you for youe helpful comments.
We improved our manuscript.
Please find our replies, point by point, in the attached file.
Kind regards.

Reviewer 2 Report
Brief summary : In this interesting opinion article (position paper), the authors are attempting to provide the readers with a perspective proposal three technological solutions: VR, telerehabilitation and assistive technology-based interventions. They also outline a background on the use of technology-aided solutions and they argue on the effectiveness and suitability of technology-mediated programs. Finally, they postulate an integrative proposal to support cognitive rehabilitation with the above technologies.
Please find included my comments.
Introduction:
The introduction is well-written and very easy to follow while conveying a vast body of information relevant to the topics of discussion.
Minor comments:
The authors state that patients with neurological impairments necessarily rely on caregivers etc. (line 32-44). This is not always the case, and the term necessarily should be used cautiously in that context.
The section on COVID-19 is not relevant to the present discussion, nor is the section about vaccine’s diffusion (Lines 51-56).
Cognitive decits and technological reports
Title error: decits should read deficits.
Lines 100-103 are missing a reference. As a matter of fact, lines 100-115 are hardly supported by the current body of literature (references should be added). As for patients assessment, in clinical practice, several health professionals might be mobilized to conduct the assessment of such patients (occupational therapists, social workers, neuropsychologists, etc.)
Lines 107: What is meant by poor resolution? Is it that it lacks quantitative data? If so, this should be explicitly stated as it is directly relevant to your discussion.
Method and selective review
To account for clarity : lines 126-136 should be its own paragraph.
Then, the comment on PRISMA or PROSPERO should be dismissed considering the article type is not a review. There is no need to justify in the methods section.
Subheadings should be used in this section to distinguish between the different steps of your review: data collection , analysis, and presentation of the main findings. The overall section is very clear and appropriately address your opinion statement.
The new perspective proposal
This section states that these technologies could be combined to enable reliable assessment and recovery of cognitive functioning in NP. However, there is no suggestion as to how it should be implemented based on the literature review. It could be argued that this is what lines 273 to 280 are attempting to do. However, this is just an oversimplification of very complex cognitive assets that cannot always be targeted by the reward system (positive reinforcers) as it is strongly correlated to the areas of the brain that have been damaged.
This section should be reworked to better convey the actual proposal that the authors are attempting to do in light of the literature review they have conducted.
Discussion
The discussion is short and could integrated examples from the literature that attempts to integrate some of these modalities. Lines 303-304 state that no empirical evidence exists. While this is the case, this can be interpreted as tautological considering that various conditions requires different assessments/treatments. Therefore, this could be further expanded as to how this proposal could solve different types of problems in NP.
Overall, this is a well written opinion piece and these are simply recommendations as to improve the presented paper.
Author Response
Dear Reviewer:
Thank you for your helpful comments.
We improved our manuscript accordingly.
Please find our replies, point by point, in the attached file.
Kind regards.
